# Primary Sclerosing Cholangitis: Burden of Disease and Mortality Using Data from the National Rare Diseases Registry in Italy

**DOI:** 10.3390/ijerph17093095

**Published:** 2020-04-29

**Authors:** Marco Carbone, Yllka Kodra, Adele Rocchetti, Valerio Manno, Giada Minelli, Alessio Gerussi, Vincenzo Ronca, Federica Malinverno, Laura Cristoferi, Annarosa Floreani, Pietro Invernizzi, Susanna Conti, Domenica Taruscio

**Affiliations:** 1Division of Gastroenterology, Centre for Autoimmune Liver Disease, European Reference Network on Hepatological Diseases (ERN RARE-LIVER), Department of Medicine and Surgery, University of Milan-Bicocca, 20126 Milan, Italy; a.gerussi@campus.unimib.it (A.G.); vincenzo.ronca@unimi.it (V.R.); federica.malinverno@unimib.it (F.M.); laura.cristoferi@unimi.it (L.C.); pietro.invernizzi@unimib.it (P.I.); 2National Center for Rare Diseases, Istituto Superiore di Sanità, 00161 Rome, Italy; yllka.kodra@iss.it (Y.K.); adele.rocchetti@iss.it (A.R.); domenica.taruscio@iss.it (D.T.); 3Service of Statistics Istituto Superiore di Sanità, 00161 Rome, Italy; valerio.manno@iss.it (V.M.); giada.minelli@iss.it (G.M.); susanna.conti@iss.it (S.C.); 4Department of Surgery, Oncology and Gastroenterology, University of Padua, 35100 Padua, Italy; annarosa.floreani@unipd.it

**Keywords:** rare diseases, primary sclerosing cholangitis, epidemiology, registry, autoimmune liver disease, cholestatic liver disease

## Abstract

*Introduction:* Studies on the epidemiology of primary sclerosing cholangitis (PSC) are mainly based on tertiary referral centers; and are retrospective case series susceptible to *selection bias*. The aim of this study was to estimate incidence; survival and cause of mortality of PSC in Italy; using population-based data. *Methods:* Data collected from the National Rare Diseases Registry (RNMR) and the National Mortality Database (NMD) were integrated and analyzed. *Results:* We identified 502 PSC incident cases. The crude incidence rate between 2012 and 2014 was 0.10 per 100,000 individuals. Sixty percent were male; mean age at disease onset and at diagnosis were 33 and 37 years; respectively; highlighting a mean diagnostic delay of 4 years. The rate of interregional mobility was 12%. Ten-year survival was 92%. In 32% of cases the cause of death was biliary-related; 12% died of biliary or gallbladder cancer. *Conclusions:* For rare diseases such as PSC; population-based cohort’s studies are of paramount importance. Incidence rates of PSC in Italy are markedly lower and survival much longer than the ones reported from tertiary; single-centre series. Moreover; the diagnostic delay and the patient interregional mobility highlights the need for increasing awareness on the disease and for resource reallocation among Italian regions within the National Health Service

## 1. Introduction

Primary sclerosing cholangitis (PSC) is a rare cholestatic disease affecting the intra and extrahepatic bile ducts that significantly affects quality of life, morbidity and mortality [1,2,3,4].

It can occur at any age and has a slight male predominance. Disease course is highly variable and there is no treatment available with proven efficacy in halting disease progression. This can result in end-stage liver failure and, as such, PSC is a leading indication for liver transplantation (LT). In Europe, PSC accounts for 9% of LT indications. Median survival time until death or LT is 12 years [3].

Most epidemiological studies on PSC are retrospective case series based on tertiary referral series with relevant selection biases [5]. Population-based studies, that include all cases in a defined geographical area, provide more accurate estimates of incidence, survival and mortality rate for the individual with PSC. Typically, multiple case-finding approaches have been used, including surveys, laboratory reports, liver histology databases, transplant registries, and death certificates. Only few population-based studies have been performed and limit the research population to a few dozen patients, reporting incidence rates ranging from 0 to 1.6 per 100,000 inhabitants. One large-scale population-based study performed in 2013 in the Netherlands reported an incidence of 0.5 per 100,000 and a point prevalence of 6.0 per 100,000 [6].

To our knowledge, there have been no epidemiologic studies in PSC carried out in Italy. We performed a population-based study using the Italian National Rare Diseases Registry to evaluate incidence, survival rate and cause of mortality of patients with PSC.

## 2. Materials and Methods

### 2.1. Data Sources and Study Population

Two data sources were used: the Italian National Rare Diseases Registry; and the National Mortality Database provided by Italy’s National Institute of Statistics.

### 2.2. Italian National Rare Diseases Registry 

The Italian National Rare Diseases Registry (RNMR), established by Ministerial Decree of 18 May 2001, n. 279, is run by the National Centre for Rare Diseases (of the National Institute of Health in Italy [7,8]. It envisaged the establishment of a network of formally designated centres (FDCs) with recognised expertise on rare diseases (RD), named the National Network of Rare Diseases. FDCs of the National Network of Rare Diseases could carry out the confirmatory diagnosis of RD, clinical assistance and treatment free of charge and ensure better effective patient management The RNMR is a scientific and institutional population-based instrument able to provide useful information for health planning, epidemiological surveillance, improvement of the governance of the National Network of Rare Diseases, both at national and regional level. The RNMR reflects the structure of the Italian National Health System, which is organized in different public bodies that work at local, regional and national levels. The local level is composed of FDCs identified in each Italian Region, which are the primary source of data flow. 

The regional authorities normally proceed to the designation of their FDC on the basis of criteria related to competence in the diagnosis, care and treatment of the RD and to the availability of appropriate complementary services. The FDCs undergo periodic (annual) monitoring by each regional authority to ensure they fulfill the criteria for designation [7,8]. The FDCs, designed at the national level, also apply to being part of the European Reference Network (ERN), implemented at the EU level. The ERN is a network connecting health care providers and centres of expertise for the purpose of improving access to diagnosis, treatment and the provision of high-quality healthcare for patients with RD. Currently, FDCs fulfilling the European criteria are officially part of specific RD ERN networks, including the ERN dedicated to rare liver diseases (ERN RARE-LIVER) [9]. 

The FDC centers “feed” the Regional Registries (RRs) with RD patients’ data (Appendix A). This is the regional intermediate level of data flow. Each RR, established by specific regional laws, communicated a common data set to RNMR to fulfill its objectives, which includes: Personal identifiers of the patient: name, surname, place and date of birth, sex, and national ID code, represented by the Fiscal Code Patient’s place of residenceName of RD diagnosed (defined according to the exemption codes listed in MD 279/2001)Name of FDC making the diagnosis, with geographical locationDate of disease onset (first evidence of jaundice, itch, or abnormal liver function tests)Date of diagnosisOrphan drugs treatment

The transmission of data is assured by the development of a bespoke IT. The IT is structured with a three-level architecture capable of connecting the FDC, RRs and RNMR according to the most rigorous legal national and European requirements for assuring the security and protection of patient personal data. The system is accessible by different users: (1) by clinicians working at FDC, who input data to RRs, (2) operators identified in each RRs who are responsible for the validation of the data of their corresponding FDCs, and (3) operators involved in the management of the RNMR at a central level responsible for the quality control at the whole databases. Every user is assigned a personal user name and password assuring different personal profiles. The data transmission process from RRs to RNMR takes place by means of a temporary channel opened for this purpose assuring that the personal data are transmitted separately from sensitive data. Personal and sensitive data are stored in separate servers of the IT service of the National Institute of Health, protected with advanced firewall and technological systems. RRs have to transmit data batches regarding confirmed diagnoses made by their FDCs to the RNMR twice a year systematically during January and July of the calendar year [7,8].

The RNMR surveys the pathologies included in the Ministerial Decree 279/2001, exempted from participation to the costs, which is not exhaustive of all RDs. The inclusion criteria to the list of RDs is characterized by rarity criteria (prevalence lower than 5 per 10,000 persons), severe clinical conditions, high level of invalidity, and costly expenses for treatment. PSC belong to the list of RDs exempted from participation to the costs, with a specific exemption code (RI0050). All cases with PSC coded with RI0050 were selected from the RNMR during the study period (1985–2014). Individuals were included at the time of their PSC diagnosis and followed up until either the end of the study period (until December 2014) or death to determine their vital status and cause of death.

The quality assurance procedure of the common data set is implemented and includes two phases. The first phase was a quality assurance procedure conducted by each RR, before sending them to the RNMR. The second phase was conducted at the central database of RNMR, after merging the data from RRs. The quality assurance assess various aspects of data quality, such as duplicate records control, missing values, inaccurate or inconsistent data values. The date of diagnosis was reported as the date of the first ascertainment of the RD diagnosis or as the date on which the diagnosis was formally certified and notified by FDCs [7,8]. In order to assess the validity of epidemiological estimates, the crude rates obtained for the whole country were compared to those deducted from the literature. 

### 2.3. The National Mortality Database 

The National Mortality Database (NMD) is run by Statistical Service Unit of the National Institute of Health and based on data provided by National Statistics Institute in Italy. The NMD cover the entire Italian Country and contains only one underlying cause of death for all persons who died from 1999 to 2014 [7,8].

ICD-10 International Classification (implemented from 2003 in the Italian health information system) is used for classifying the causes of deaths. The K83.0 from ICD-10 was used to codify mortality cases with PSC. ICD-10. This code is not specific for PSC as it identifies a large group of cholangitis patients, which includes other specific diseases as follows: Cholangitis NOS, ascending; primary; recurrent; sclerosing; secondary; stenosing; suppurative and it is not possible to distinguish PSC from other types of cholangitis. For all mortality cases, the causes of death were analyzed and presented by major disease categories of the ICD-10 classification.

### 2.4. Record Linkage between Italian National Rare Diseases Registry and the National Mortality Database

A deterministic linkage between RNMR and NMD was achieved to evaluate the vital status and causes of death of cases included in NMD; being performed by the Unit of Statistics of the National Institute of Health, in the framework of the Italian National Statistics Program, who provide official statistical information of public interest at the national level. The above-mentioned linkage is compliant with the General Data Protection Regulation (EU 2016/679) and its national application by the Legislative Decree 10/8/2018, n.101.

The linkage methods were performed in two steps: the first linkage key used was the fiscal code variable (Italian administrative identification code); in case the fiscal code was missing in NMD a second key linkage used was constituted by the name, surname and date of birth.

In order to study the patients’ life status, the variable date of diagnosis was taken into account because, at that time, patients were certainly alive.

### 2.5. Diagnosis of PSC and Study Definitions

The diagnosis of PSC is clinical and based on specific findings at magnetic resonance cholangiopancreatography (MRCP), along with abnormal liver function tests. To reach a diagnosis of PSC, the clinician should exclude potential causes of secondary sclerosing cholangitis (trauma, surgery, ischemia, etc.). Once the diagnosis is made, the patient is granted an exemption code (RI0050) specific for the disease, which allows the patient cover for health-related expenses for that specific disease. All patients with code RI0050 have PSC.

We defined an incident case as a patient who had been newly diagnosed with PSC in the RNMR during a given year. In the survival analysis, the study entry was the time of diagnosis of PSC. The endpoint was death. Individuals who did not reach the event “death”, were censored at the date of database lock (31 December 2014).

The diagnostic delay was defined as the time from “date of initial symptoms of PSC” until the “date of diagnosis”.

### 2.6. Statistical Analysis

The incidence was calculated by dividing the total number of incident cases by the entire Italian population for a given year. The crude incidence rate of PSC was analyzed between 2012 and 2014, because the RNMR reached the full national coverage in 2012. The ANOVA model was used to compare the mean age at diagnosis, mean age at onset and diagnostic delay in males and females.

During the study period (1985–2014), the probability of survival and 95% confidence interval (CI) were estimated according to the Kaplan–Meier method. The log-rank test was used to compare the probability of survival between genders. Inter-regional mobility was determined as the rate of patients moving from their resident region to another region where the patient received care.

Statistical analyses were performed using SPSS 24.0 and SAS software.

## 3. Results

### 3.1. Patient’s Characteristics

During the study period (1985–2014), a total of 502 incident cases of PSC were identified in Italy, with a population of 60,795,612 (31 December 2014, Italian National Statistics Institute source). Clinicians included PSC cases retrospectively diagnosed since 1985, before the date of establishment by law of the RNMR in 2001. The vital status could not have been evaluated for all PSC cases as the national mortality database covers a period from 1999 to 2014.

The main demographics and clinical characteristics of the patients with PSC are reported in Table 1. Briefly, 302 (60.2%) were men with a male to female ratio of 1.5:1. Mean age at disease onset was 32.7 years (SD = 17.1) and mean age at diagnosis was 36.9 years (SD = 17.5). The mean diagnostic delay was 4.0 years (SD = 5.6). Based on quartile distribution, 75% of cases had 5.9 years and 30% had 0.33 years of diagnostic delay. There were no statistically significant differences in the age at diagnosis, age at onset and diagnostic delay among male and female (*p* > 0.005).

### 3.2. Incidence 

The crude annual incidence rate of PSC in 2012–2014 is shown in Figure 1, Table 2 and Appendix A**.** The mean number of new patients with PSC registered annually between 2012 and 2014 was 62. Based on the resident registration data from the Italian National Statistics Institute, the mean calculated crude annual incidence was 0.10 per 100,000 persons (95% CI: 0.08–0.13). Crude incidence did not change during the observation period going from 0.09 (95% CI: 0.68–0.11) in 2012 to 0.1 (95% CI: 0.07–0.12) per 100,000 population in 2014. The age specific incidence rates are show in Table 3.

### 3.3. Mortality

Out of 502 patients, 460 were alive on 31 December 2014 and 25 had died. For 17 of the 502 patients, the vital status could not be determined. During the study, there were 25 deaths, of which 12 (48%) were PSC-related. Characteristics of individuals who had died and their cause of death are shown in Appendix A. Briefly, “cholangitis” was listed as an underlying cause of death in 8 cases (32%); cancer of the gallbladder or the biliary tract was listed as an underlying cause of death in 3 cases (12%). All deaths were adults aged from 35 to 86 years.

When including all causes of deaths in the analysis, the survival rate was 92% at 10 years from diagnosis and 82% at 20 years. The Kaplan–Meier curves shown no significant difference between male and female on estimated survival times from diagnosis (*p*-value obtained from the log-rank test method = 0.67) (Figure 2).

### 3.4. Interregional Mobility

We observed that 61 patients (12%) of those who received their exemption code for health coverage free of charge moved to regions different to the one where they live.

## 4. Discussion

Most epidemiological studies on PSC have been performed in tertiary referral centers, which make them prone to selection bias and unable to obtain accurate epidemiological data [4]. Only six high-quality population-based studies have been performed so far describing only a few dozen patients for each study and reporting incidence rates ranging from 0 to 1.6 per 100,000 inhabitants. The largest study from the Netherlands described a cohort of 590 PSC patients identified from 44 hospitals in a large geographically defined area of the Netherlands, comprising 50% of the population (≅8 million) [6].

In our study, we present one of the largest population-based studies of PSC, conducted using the Italian National Rare Diseases Registry. The cohort includes 502 PSC patients identified at a nationwide level in the entire Italian population of ≅60 million. We report an incidence rate in the period 2012–2014 of 0.10 per 100,000 persons, which remained stable during the study period. The highest incidence of PSC was observed in male individuals and in the age group of 41–50 years.

In our study, the incidence rates of PSC were lower compared to those reported in other population-based cohorts in Europe. In more detail, in 2010, Lindkvist et al. reported the epidemiology of PSC in an adult population in Vastra Gotaland, a region in southern Sweden with a defined population of about 1.5 million, in the time-period 1992–2005 [10]. They identified 199 incident cases of PSC and reported an annual incidence of 1.22 per 100,000 in the total adult population. In order to find cases, they used inpatient and outpatient registers at all departments of internal medicine and surgery at all hospitals in the region using a computerized search for relevant codes according to the ICD9 and ICD-10 version (codes 576 and K830, respectively). In 2008, a UK study based on the General Practice Research Database conducted between 1991 and 2001 identified 149 incident cases and reported an incidence of PSC of 0.41 per 100,000 person years [11]. In 2017, Liang et al. identified 250 incident PSC patients using the UK Clinical Practice Research Datalink. The age-standardized incidence of PSC was 0.68 per 100,000 person-years and the age-standardized prevalence was 5.58 per 100,000 during 1998 to 2014 [12]. In 2013, Boonstra et al. [6] reported a Dutch cohort of 590 PSC patients, resulting in an incidence of 0.5 per 100,000. They used four independent hospital databases: a nationwide network and comprehensive registry of histopathology and cytopathology in the Netherlands; a hospital billing system; endoscopic retrograde cholangiography reports in endoscopy-suite databases; and personal lists of treating physicians. The study enrolled individuals in 44 hospitals from 2000 onward in a geographically defined area of six adjacent provinces comprising 50% of the Dutch population. 

Our data are in keeping with those reported by Escorsell et al. in Spain in 1994 [13]. They surveyed 33 hospitals throughout Spain to ascertain the number of PSC patients from January 1, 1984 to December 31, 1988. Twenty-three centers, from 12 Spanish regions, covering a population of 19.23 million answered the questionnaire. They identified 43 patients with an overall annual incidence of 0.07 per 100,000.

The estimated median survival until LT or PSC-related death found in the present study is much longer compared to previous reports. We analyzed death from any cause as endpoint for Kaplan–Meier analysis. The survival rate was 92% at 10 years and 82% at 20 years. These figures are definitely better than the ones reported in studies, including tertiary referral cohorts, of approximately 60% at 10 years, but also of the Dutch population-based study, which reported a 10-year survival of approximately 80%. Since there are no therapeutic approaches, this difference might suggest a different disease phenotype or natural history. In our study, the disease has a male predominance, though it can affect men and women.

Other than epidemiological hints, population-based registry data can also provide a comprehensive framework for evaluating nationwide variations in healthcare infrastructure on the diagnostics and health outcomes of rare diseases such as PSC. We observed a heterogeneous distribution of PSC incidence by region ranging from 0.2 per 100,000 in Basilicata (576,619 inhabitants at 31 December 2014) to 3.7 per 100,000 in Provincia Autonoma di Bolzano (518,518 inhabitants at 31 December 2014). Differences in geographical incidence must be influenced by various factors: geographical coverage of the registry and the under-reporting rate of data being the most plausible; however, biological or environmental components cannot be excluded.

We highlight a mean diagnostic delay of 4 years. Of note, there was no change in the use of diagnostic tools during the study period. There is an open debate on how to improve the diagnostic delay for patients with rare diseases, which is on average 4.8 years [14]. For many patients with rare diseases, their ‘diagnostic odyssey’ might be partly related to a delayed referral from primary care physicians, especially if patients present with common symptoms like fatigue and itch; more importantly, multiple clinical centres, specialties, and investigations are involved after the referral.

We described interregional mobility, with 12% of patients overall moving to other regions. Patient interregional mobility is a phenomenon with important social and economic consequences and concerns social and territorial equity, resource reallocation among regions as well as the role of central and regional government within a National Health Service system. Such phenomena might be even more common for rare diseases, where only a few specialized centres are available on the national territory.

A strength of the study is the use of a potential tool to capture rare disease cases at a national level, such as in a population-based registry. For this, it is essential to include all existing patient cases because completeness is of critical importance. The Italian Registry is the first population-based registry implemented in Europe and it is a useful tool for generating health indicators relating to a considerable number of rare diseases, rather than to specific conditions. At a Europen level, only a few population-level registries of rare diseases are currently available. The NRRD is now implemented in all the Italian Regions with considerable variation of incidence across the Italian territory but data quality and completeness still need to be improved.

Certain possible methodological limitations of the study have to be taken into consideration. In the present study, we used the unique, disease-specific exemption code RI0050 for patient identification. The reliability of exemption codes primarily depends on the coding requirements imposed on the providers by the healthcare system. Since detailed coding for exemption code RI0050 is a prerequisite for reimbursement for all diagnostic procedures and intervention specifically for PSC, the occurrence of a single code for a disease can be considered sufficient and specific to identify a patient. Even with a powerful tool such as a population-based national registry on rare diseases, there is always a chance of missing some cases and therefore slightly underestimating incidence rate. This might be related to an underperformance of each FDC in reporting the new cases at each regional registry. Obviously, some cases may have been missed, because the early stages of PSC may be without any symptoms.

In addition, since inflammatory bowel diseases are not included with a specific code to the RNMR, we could not explore the association of inflammatory bowel diseases with PSC. Finally, considering the limited study period analysed in our work, and based on the difficult identification of the best timing for LT in PSC and the heterogeneous application of extra-MELD points in PSC in clinical practice, carrying out a thorough comparison with epidemiologic data from others countries is not straightforward.

## 5. Conclusions 

We provide the first estimates of incidence of PSC in Italy, which are markedly lower than those previously reported, mainly from the Nordic countries. This shows that differences in the disease burden might be true, and might be related to a reported North-South gradient of autoimmune liver conditions. We should however, be able to have solid comparisons between studies based on uniform case-finding and diagnostic criteria that may overcome the possible limitations caused by different degrees of awareness, particularly for rare conditions. We reported diagnostic delay and interregional mobility, which represent a challenge for the healthcare system. Finally, we reported a longer survival compared to cohorts from tertiary referral centers, exemplifying the effect of selection bias, but also compared to population-based cohorts, which might suggest a different disease phenotype or disease course. Extensive population-based studies are needed to accurately establish epidemiology and disease course in patients with PSC.

## Figures and Tables

**Figure 1 ijerph-17-03095-f001:**
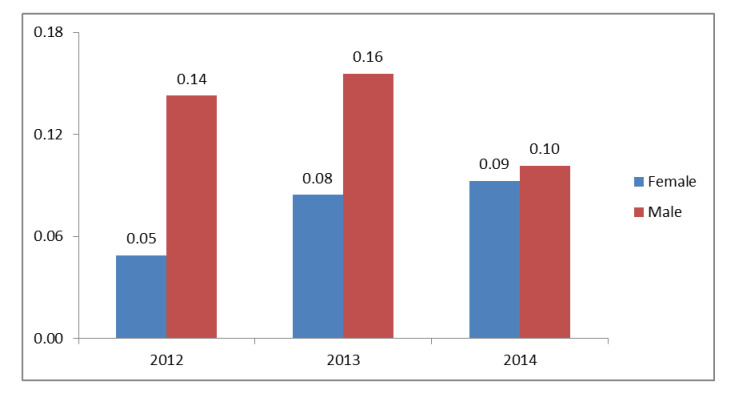
Crude incidence of primary sclerosing cholangitis per 100,000 individuals in the years 2012–2014 stratified by gender.

**Figure 2 ijerph-17-03095-f002:**
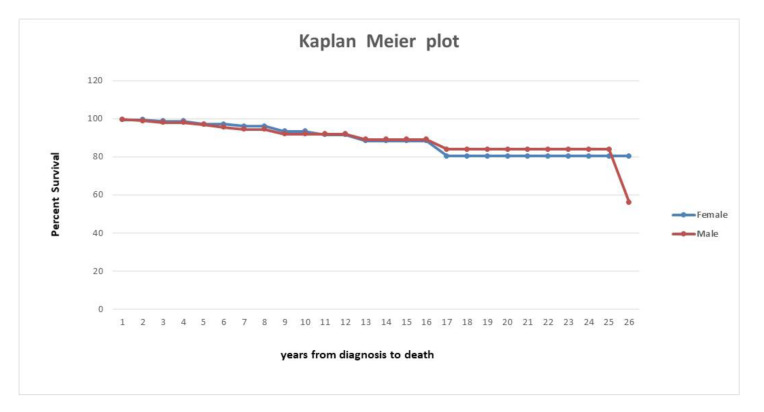
Survival of patients with PSC from the time of diagnosis stratified by sex (n = 485).

**Table 1 ijerph-17-03095-t001:** Demographics and clinical characteristics of individuals with primary sclerosing cholangitis (1985–2014) registered in the Italian National Rare Diseases Registry.

	Living	Deceased	Total
**Sample**	477	25	502
**Sex (M:F)**	1.5:1	1.8:1	1.5:1
M	286	16	302
F	191	9	200
**Peadiatric (0–17)**	45	0	45
**Adult (18–65)**	381	16	397
**Eldery (>65)**	50	9	59
**Age at enrollment (average)**	42.7	61.4	43.6
**Age at diagnosis (years)**			
Average	36.9	53.2	37.7
Median	38.9	49.0	39.7
Standard Deviation	17.5	15.5	17.8
**Age at initial symptom (years)**			
Average	32.7	48.5	33.5
Median	32.8	44.9	33.8
Standard Deviation	17.1	20.4	17.6
**Diagnostic delay (years)**			
Average	4.0	3.1	4.0
Median	1.2	0.3	1.2
Standard Deviation	5.6	6.2	5.6

**Table 2 ijerph-17-03095-t002:** Incidence cases with primary sclerosing cholangitis and crude incidence rate per 100,000 individuals from 2012 to 2014.

Year	Incidence Cases with PSC	Total Population	Crude Incidence Rate per 100.000 Individuals	95% CI
2012	56	59685227	0.09	0.68–0.11
2013	71	60782668	0.12	0.88–0.14
2014	59	60795612	0.10	0.07–0.12
2012–2014	186	60421169	0.10	0.08–0.13

**Table 3 ijerph-17-03095-t003:** Incidence rate with primary sclerosing cholangitis and age specific incidence rate per 100,000 individuals from 2012 to 2014.

Age Groups	2012	2013	2014	Total Incidence Cases	Population Average (2012–2014)	Incidence Rate /2012–2014)
0–10 years	3	2	2	7	18,360,804	0.04
11–20 years	3	3	1	7	17,131,081	0.04
21–30 years	8	5	12	25	17,325,974	0.14
31–40 years	11	14	6	31	25,149,480	0.12
41–50 years	19	21	16	56	29,103,037	0.19
51–60 years	7	12	11	30	24,163,079	0.12
61–70 years	2	10	7	19	20,617,527	0.09
71–80 years	2	4	3	9	16,497,870	0.05
80 +	1	0	0	1	10,110,768	0.01
Total	56	71	58	185	17,8459,620	0.10

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
