# Peer review of "Primary Sclerosing Cholangitis: Burden of Disease and Mortality Using Data from the National Rare Diseases Registry in Italy"

_ijerph, 2020, doi:10.3390/ijerph17093095_

Round 1
Reviewer 1 Report
Materials and methods
Please define how FDC have been designated and by whom.
Is the designation subject to periodic monitoring, based on which criteria?
Which is the relationship between ‘FDC’ and European Reference Networks, in particular ERN Liver?
Please describe how data are transferred from the FDC to the regional registries and to the national registry, particularly as patients’ data are not anonymized.
Authors briefly describe the record linkage process performed, based on identifiable patients’ data. Please explain why it is not based on non-identifiable patients’ data. How this process can be defined as compliant with the General Data Protection Regulation in force and to its national application, that could not certainly be less restrictive than the GDPR? In addition, the issue of patient consent is not addressed by Authors. Please specify how patients’ consent is obtained.
Date of disease onset. This is a very interesting variable to be collected by a registry on PSC. But to evaluate its quality Authors could better explain how it is collected: by clinicians, by patients, by both?
How disease onset is established? Please specify if this information is based on laboratory exams, on other clinical documentation.
If not, Authors should be careful when considering data on the diagnostic delay, as they could be affected by bias (to be mentioned in the discussion as a limit of the study, if appropriate)
Diagnosis
I think Authors should better explain how cases are enrolled in the registry.
It seems that the registry is organized as an administrative database as Authors mention the use of an exemption code to define cases. So, for the reader it is not clear if it is a clinical registry or not, and if yes, to what extent. It could be very useful for readers to know which criteria have been adopted, providing the appropriate reference/s. Please specify also the definition of exclusion criteria i.e. causes of secondary sclerosing cholangitis. This is a very important point to be addressed.
RESULTS
When presenting the results, it should be clarified which is the study period considered (depending on the source of data: PSC registry vs national mortality database).
If the PSC registry within the national RD one, has been stablished in 2000, then the statement “ During the study period (1985-2014), a total of 502 incident cases of PSC were identified in Italy in a population of 60.795,612 (line 135-136) is confusing. The study period to which data are referred should be added also in Table 1.
Results are presented as descriptive. So, I think lines 141-142 can be considered non informative and can be deleted.
It could be very interesting if Authors could enrich the results section with age adjusted incidence and prevalence rates.
Table 1
Some minor errors should be corrected:
Not mediana, but median
Add “years” where appropriate i.e. Age at diagnosis, diagnostic delay, etc…
Interregional mobility
12% does not represent a high mobility rate, considering that PSC is a rare disease and requires highly specialized care. Information on % without reporting the number of cases experiencing migration for care can be misleading and is not of interest for the majority of potential readers, not familiar with “national” issues.
Supplementary tables
Table 1
I have severe doubts in reporting low number of cases that associated with the place of residence can make them easily identifiable.
Looking at the numbers, some figures are striking: which is the possible explanation for “Pulia” having 1 case in 2012, 24 cases in 2013 and 6 reported in 2014?
Reporting rates are very heterogenous, not only during years (only three considered), but also considering similar denominators (similar population sizes). How this can be explained and affects national results?
Discussion
Population coverage
Please address the issue of registry completeness. As it is a population-based registry, potential limits leading to underestimation of cases should be mentioned in the discussion section.
Some Authors reported an increase in PSC incidence, even though it is not clear whether these data reflect a true increase in disease occurrence or a better disease detection. The study period considered in this article is too limited to compare presented data with others coming from Northern European countries. As Authors state that the registry has been stablished in 2000, it would be very interesting to consider an extended observation period.
Patients with PSC have good outcomes after liver transplantation compared to other indications. Differences in clinical practice regarding patients eligibility to liver transplantation exist between countries, i.e. the US vs European countries. Please discuss if and how this can affect survival data in your study.
In general, potential limits and strengths of the study could be better addressed in the discussion section.
Bibliography
It could be enriched with other articles.
At least the following should be added:
Dyson JK et al. Primary sclerosing cholangitis. Lancet. (2018)
Lazaridis KN et al. Primary Sclerosing Cholangitis. N Engl J Med. (2016)
Karlsen TH et al. Primary sclerosing cholangitis - a comprehensive review. J Hepatol. (2017)
Author Response
Please find author's reply attached.

Reviewer 2 Report
For rare diseases such as PSC, population- based cohort’s studies are of paramount importance.
Population-based studies, that include all cases in a defined geographical area, provide more accurate estimates of incidence, survival and mortality rate in compare to retrospective studies based on tertiary source.
Can you explain the background with which you included the patients at 14 as adults?
Did you have a jurisdiction from the local ethics committee?
In the table 1 please correct the age groups please correct the age groups. If the first children group ends with 14, then the second group adult should start at 15, similar with the next group.
Figure 2 should receive a correct title, not a name of the statistical description.
Do you have observed regional differences between north, south, centre of Italy and islands?
It would be very interesting to analyze comorbidities with regard to PSC in your population.
Author Response
Please find author's reply attached
